# Overcoming both Domain Shift and Label Shift for Referring Video Segmentation

## ABSTRACT

Open-set domain generalization (OSDG) aims to enhance the robustness of the model when facing both domain shift and label shift, highlighting a wide range of potential in real-world applications. However, previous OSDG methods can only recognize seen objects and mark all unseen objects as "unknown" categories during inference, which is far from satisfactory. In this paper, we explore the scenario of referring video segmentation to study how to make the model maintain good segmentation ability for unknown objects under OSDG setting. To bridge the huge gap caused by label shift, we propose CLIP-based Reasoning Prompt (CRPrompt), which can combine text and visual prompts together to improve text-object matching ability of CLIP, transferring the segmentation ability to unseen classes based on the knowledge learned from seen classes and large-scale text-image pairs, i.e., color, shape, spatial relationships. Meanwhile, to improve the robustness of CRPrompt, we propose Retrieval-augmented Instance Normalization (RaIN), which can effectively enhance the robustness of the model by retrieving visual objects with similar semantic concepts through input query and performing Instance Norm among them. Extensive experiments on open-set and zero-shot domain generalization tasks demonstrate the effectiveness of our approach. The code is available in the supplementary material.

## 1 INTRODUCTION

As one of the most important visual-language understanding tasks, question-based video segmentationGavrilyuk et al. (2018); Wang et al. (2019; 2020) has been widely studied, aiming to predict pixel-level masks for actors or objects in videos based on the given natural language query. Nevertheless, these works are all based on large-scale manual labeling, human annotators must label every possible pixel in thousands of frames. Moreover, real-world applications present a highly complex scenario: models trained on the source domain may not only encounter distribution deviations in the target domain (domain generalization problem) but also face previously unseen categories (open-set problem). It is very unrealistic to rely on human annotation in every new situation. As we can see in Figure 1 (a), compared with the source domain (A2D dataset), target domain (RVOS dataset) contains novel nouns and objects, i.e., "frisbee". And the visual features from different domains also have domain shifts in many aspects, i.e., background, lighting, appearance, and action of actors. These kinds of label shift and domain shift will dramatically decrease the segmentation ability of the model, as shown in Figure 1 (b).

To mitigate the aforementioned issues, two novel paradigms have been introduced: open-set domain adaptation Bucci et al. (2020); Jang et al. (2022) and open-set domain generalization Zhu & Li (2021); Yang et al. (2022). These works segregate the target-unknown features while only aligning the source and target-known distribution, successfully separating seen and unseen classes in target domains. However, there is still an intrinsic limitation of these methods, that is, all unseen objects will be marked as "unknown" category, which obviously cannot meet the requirements of referring segmentation task. In this paper, our objective is to develop a model that is not only robust enough to resist domain shifts but also capable of maintaining high segmentation performance when encountering new categories.

Previous works Wang et al. (2022); Zhong et al. (2022) use CLIP's Radford et al. (2021) powerful text-image pairing capabilities to match words with image regions, which can achieve zero-shot segmentation results. However, these methods are over-dependent on the pre-trained CLIP model,

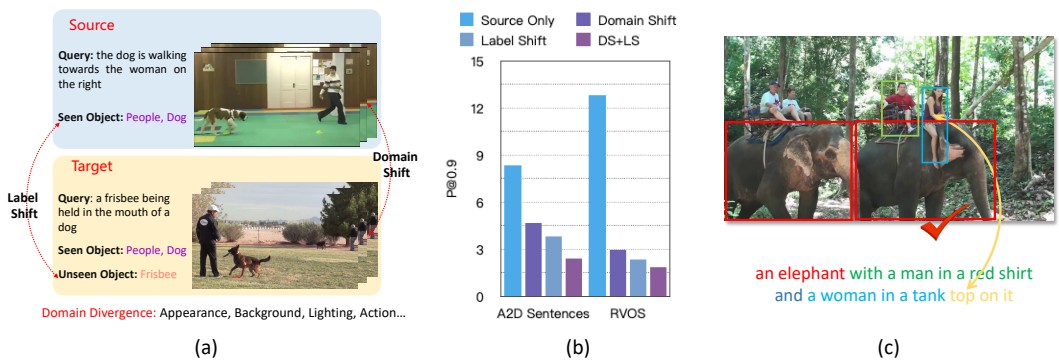

Figure 1: (a) The illustration of the label shift and domain shift in our task. (b) The necessity of introducing OSDG methods (left: A2R, right: R2A). As we can see, the performance of the model drops a lot under domain shift and label shift. (c) The illustration of our main idea in this work. Even though the object "elephant" is unknown, the model can align the known text-object pairs together, combining these clues and spatial relationships to localize the right region of the described elephant.

yielding several shortcomings: 1) CLIP focuses more on global information, while the referring video segmentation requires fine-grained interaction between visual regions and text fragments. 2) These approaches overlook the use of relationships among visual regions and reference contexts, are unable to judge unseen categories. 3) Furthermore, these CLIP-based methods do not consider the performance degradation caused by domain shifts.

For humans, a common practice is to combine the spatial relationships described in the reference and the visual concepts that have seen before together, to reason about which object region is the unfamiliar noun in the question refers to. As we can see in Figure 1 (c), although the model has not seen "elephant" before, it still can use the known visual concepts "man" and "woman", and the learned text-visual knowledge like "red" and "top on it", to localize the right elephant. This process of using existing knowledge as a prompt is similar to the prompt learning in NLP Lester et al. (2021); Shin et al. (2020), and has also made progress in vision-language tasks Zhou et al. (2022a); Yao et al. (2021). Inspired by this, we develop CLIP-based Reasoning Prompt (CRPrompt), which combines CLIP and prompt engineering together to reason unseen object categories based on knowledge learned from seen categories, effectively mitigating label shift problems. We design two different reasoning ways to train the text and visual prompts, **implicit reasoning** and **explicit reasoning**. Thereby, visual concepts, text information and spatial logical relationships can be aligned together in the latent space. When facing new object categories during test, the model can make use of the known objects around that have been seen in source domains, or the learned knowledge, i.e., color, shape, and spatial relationships, to predict the right regions.

To further improve the robustness of the model to resist domain shifts, we design a novel module named Retrieval-augmented Instance Normalization (RaIN). RaIN can retrieve several objects similar to the current visual concept from a dynamically updated memory based on the reference text. Then we extract the statistical style from these similar visual features to update the current object, simulating the different styles that the current object may have in the unknown domain. Compared with traditional Adaptive Instance Normalization (AdaIN) Huang & Belongie (2017); Peng et al. (2022) that can only update the feature style within the same batch, our RaIN can introduce arbitrary external knowledge to apply meaningful data augmentation. To be summarized, our contribution is threefold:

- To the best of our knowledge, this work is the first attempt to solve open-set domain generalization problem on multi-modal task, which is fundamentally more challenging.

- We bring up a novel framework named CRPrompt, which can make full use of learned visual-language cues to predict unknown objects. Also our proposed RaIN can improve the robustness of the main model against domain distribution divergence.

- The extensive experiments on open-set and zero-shot domain generalization tasks demonstrate that our methods can effectively mitigate the performance degradation problem caused by label and domain shift.

## 2 PRELIMINARY AND RELATED WORK

### 2.1 CLIP FOR SEMANTIC SEGMENTATION

CLIP consists of two parallel encoders, the vision encoder $e_{img}$ transforms an image into a visual embedding feature $e_{img}(I_k) \in \mathbb{R}^D$, the text encoder does the same thing for the given text $e_{txt}(T_k) \in \mathbb{R}^D$, where $D$ is the feature dimension, $(I_k, T_k)$ is the paired image and text. CLIP is trained with a contrastive loss to align the paired image and text in the feature space. By training on 400 million various text-image data, CLIP shows incredible achievements in zero-shot image classification ability.

CRIS Wang et al. (2022) modifies the vanilla CLIP, enabling it with text-to-pixel interaction ability however, it lacks the zero-shot segmentation ability. Lüddecke & Ecker (2022) design three different image operations which can improve the alignment between the object and text prompts. RefCLIP Subramanian et al. (2022) employs a semantic tree to analyze the spatial relationships of phrases in the queries, then use these phrases to predict the corresponding regions of the images. These methods Lüddecke & Ecker (2022); Subramanian et al. (2022) have proven zero-shot ability in combination with CLIP in referring image segmentation tasks.

### 2.2 VISION-LANGUAGE PROMPT

CLIP first introduces prompt learning in vision-language tasks, by designing the text prompt "A photo of a {CLASS}.", where "{CLASS}" can be replaced by the current object category. These can be directly adapted to downstream tasks without fine-tuning. The past few years have witnessed tremendous progress in the vision-language prompt area Yao et al. (2021); Jia et al. (2022); Du et al. (2022); Shen et al. (2022). CoOp Zhou et al. (2022b) introduce M learnable vectors $\{v_1, v_2, ..., v_M\}$ that can be end-to-end learned from given visual data to replace the hand-crafted text prompt used in CLIP. However, CoOp cannot be generalized well to unseen classes. CoCoOp Zhou et al. (2022a) improves the generalization ability of CoOp by replacing the fixed prompt tokens with instance-specific context tokens, which can efficiently alleviate the overfitting problem. Specifically, they apply a lightweight neural network $h_\theta(x)$ to generate a conditional vector $\phi$ for each input visual feature, and then combine it with M learnable vectors together: $t_i(x) = \{v_1(x), v_2(x), ..., v_M(x), c_i\}$, where $t_i(x)$ is the prompt for i-th class, $c_i$ is the word embedding for i-th class name.

In this work, we follow CoCoOp to design our vision-language prompt, which can promote the generalization ability of the model when facing unknown categories.

### 2.3 OPEN-SET SINGLE DOMAIN GENERALIZATION

Single domain generalization aims to improve the model's generalization ability learned from just one single source domain. Previous works only assume the label spaces of source and target domains are the same Wang et al. (2021); Qiao et al. (2020), while in realistic scenarios, different scenes often contain different object categories. To alleviate this problem, Zhu & Li (2021) bring up a new setting, open-set single domain generalization (OS-SDG). They introduce a CrossMatch approach to generate auxiliary unknown class samples with adversarial data augmentation strategy. Yang et al. (2022) think the unknown class is the closest class to any known class. Thus they propose an (n+1)-way classifier for n known classes where the second-largest prediction score is for unknown class. These works aim to separate the unseen objects from the seen objects during test, only classifying the seen objects. However, in this work, we hope that the model can not only recognize the unknown categories but also accurately segment them, even though the target word in the query and the object in the frame have not been learned before.

## 3 METHOD

### 3.1 PROBLEM SETTING AND MODEL OVERVIEW

First we give a detailed description of referring video segmentation problem: given a natural language query $Q$ and its counterpart video frames $V$, the model is required to generate accurate segment masks on the objects related to the input query. Under open-set single domain generalization setting, the model should improve its generalization ability against both domain shift and label space shift on

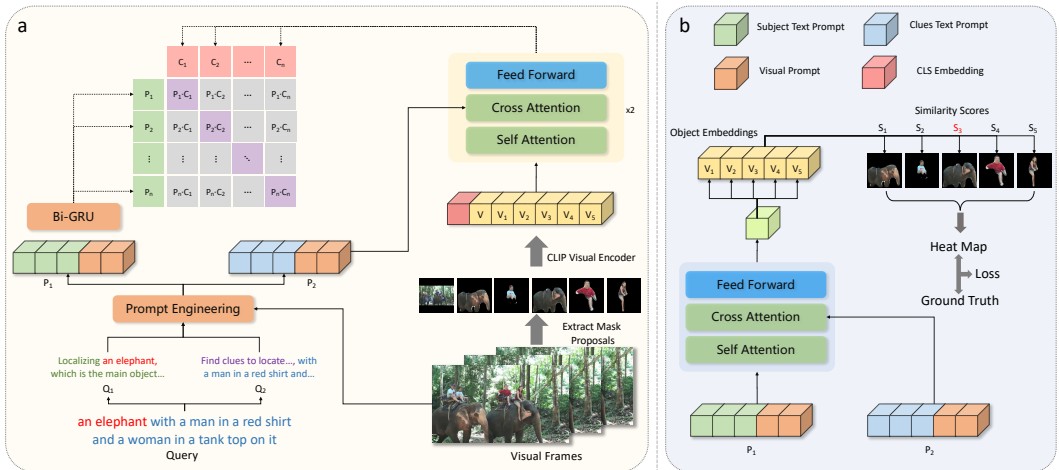

Figure 2: The overview of our proposed CRPrompt. We first divide the given query $Q$ into two parts: the subject part $Q_1$ to be identified, and the remaining part $Q_2$ that describes $Q_1$. We use CoCoOp to perform prompt engineering on $Q_1$ and $Q_2$, and then design two kinds of reasoning: (a) in the implicit reasoning, we incorporate $Q_2$ into pre-extracted class-agnostic mask proposals to perform contrastive predict learning with the subject part $Q_1$; (b) in the explicit reasoning, we select the most matching proposal corresponding to the subject, then integrate the selected proposal into the backbone network.

unseen target domains $\mathcal{T} \in \{\mathcal{T}_1, ...\mathcal{T}_W\}$ with the segmentation model trained only on single source domain $\mathcal{S}$. The label space $C_t$ of the target domain has novel classes that do not belong to the source domain $C_s$, $C_t^u = C_t \backslash C_s, C_t^u \neq C_t$. Following Zhu & Li (2021), we formulate the worst-case problem in our problem as follows:

$$\min_{\phi} \sup \mathbb{E}[\mathcal{L}_{\text{seg}}(\phi; Q_k, V_k; \mathcal{T}_k) + \mathcal{L}_{\text{seg}}(\phi; Q_u, V_u; \mathcal{T}_u) : D(\mathcal{S}, \mathcal{T}_k) \leq \rho_1, D(\mathcal{T}_k, \mathcal{T}_u) \leq \rho_2], \quad (1)$$

where $Q_k, V_k$ are query and visual representations in known classes domain $\mathcal{T}_k$; $Q_u, V_u$ are query and visual representations in unknown classes domain $\mathcal{T}_u$. To bridge this gap, we propose a CRPrompt to transfer the segmentation ability to unseen classes against label shift based on the knowledge learned from seen classes and large-scale text-image pairs used in pre-trained CLIP model(Section 3.3), and design a RaIN to improve the robustness of the model to against domain shift(Section 3.4).

## 3.2 BASELINE MODEL

Different from previous work using Maskformer Cheng et al. (2021) as a backbone to segment all objects in the image, we need to locate the specific object that matches the text description in the video. Therefore, following with previous work Hui et al. (2021), we use an I3D layer Carreira & Zisserman (2017) as our encoder and decoder, we divide the visual features in the encoder and decoder layers into K different scales respectively, denoted as $\hat{V}_k \in R^{H_k \times W_k \times C}$, where $H_k$, $W_k$ and $C$ are height, width and channel number of the i-th visual feature, respectively. We use Bert to encode question as $\hat{Q}$, since Bert can extract semantic features more robustly when encoding texts with different styles from different domains. We use the visual-language cross-attention network on each visual scale following Wang et al. (2019) to achieve cross-modal interaction. Besides, to reduce the gap between seen and unseen classes, we extract $M$ mask proposals $\{v_1, v_2, ..., v_m\}$ for each video using pre-trained models such as Mask-RCNNHe et al. (2017), then we use our proposed CRPrompt to select the most relevant proposal from these mask proposals, and integrate it into the main segmentation backbone network.

## 3.3 CLIP-BASED REASONING PROMPT

**Prompt Engineering.** Prompt is an ideal handle to exhaustively leverage the pre-trained knowledge of CLIP, which has been proven useful for boosting CLIP's generalization ability in a zero-shot

manner Shu et al.; Zhou et al. (2022a). In this work, we try to combine language and visual cues together to design prompts for input questions. Specifically, given a natural language query $Q$, which consists of two parts, the subject part $Q_1$ to be located and the remaining part $Q_2$ describing $Q_1$. Following Subramanian et al. (2022), we use spaCy Honnibal & Johnson (2015) to split the input reference into several independent phrase fragments, the subject part $Q_1$ is always at the beginning, then we connect the following descriptive phrases with "and" as $Q_2$ (see Appendix for more details). We introduce text prompts and visual prompts in $Q_1$ and $Q_2$, respectively:

$$P_1 = \text{CLIP}_t[L_1; Q_1; L_2] \oplus P_v; \quad P_2 = \text{CLIP}_t[L_3; Q_2] \oplus P_v, \tag{2}$$

where $L_1$ is *"Localizing "*, $L_2$ is *", which is the main object to be segmented."*, $L_3$ is *"Find clues to locate the previously mentioned subject from the following objects and relationships, "*, $P_v$ is the visual prompt generated from video frame features using CoCoOp. The natural language prompt can let the model know which phrase is the subject to be recognized and which phrases can provide useful clues to help identify the subject. The visual prompt can allow the model to dynamically adjust the prompt according to the visual content, improving the generalization ability to unknown categories.

**Reasoning.** We design two kinds of reasoning to fine-tune the prompt, explicit reasoning and implicit reasoning. For the **implicit reasoning**, we first resize the visual frames and $M$ mask proposals to CLIP resolution, and connect them together with a $[CLS]$ token: $\hat{V} = \{v_{cls}, v, v_1, v_2, ...v_m\}$. The introduction of the entire video frame embedding $v$ can facilitate the model to get a holistic understanding of the spatial relationship among these object proposals. We feed $P_2$ and $\hat{V}$ into an Encoder layer that maps text-visual modalities to a shared semantic space, which consists of two layers of cross-attention and self-attention (see Figure 2 for more details). We use a Bidirectional GRU to compress $P_1$ as $p \in R^D$, and choose $v_{cls}$ as an overall representation of visual contents, then we use $v_{cls}$ to predict $p$:

$$\mathcal{L}_{\text{const}} = -log \frac{exp(sim(v_{cls} \cdot p_i)/\tau)}{\sum_{n=0}^{N} exp(sim(v_{cls} \cdot p_n)/\tau)}, \tag{3}$$

where $sim(,)$ denotes the cosine similarity, $N$ is the batch size, $\tau$ is the temperature of the softmax function. In this way, the knowledge related to the main object will be extracted from $P_2$ and $\hat{V}$ to predict the right subject, which in turn makes the model prone to find the most relevant mask proposal through $P_1$ during explicit reasoning.

In the **explicit reasoning stage**, the model needs to select the most matching mask proposal according to the subject, then incorporate it into the main segmentation network as auxiliary information, which is also an important step to achieve zero-shot referring segmentation ability. We first integrate the information related to the subject from $P_2$ into $P_1$ with cross-attention, such as surrounding objects, spatial relationships, etc. Therefore, the model has a more comprehensive understanding of the context information around the subject, promoting the performance when predicting the corresponding object proposal. Then we use Bidirectional GRU to compress $P_1$ as $\hat{p}$, and compute the similarity scores $S = \{s_1, s_2, ..., s_M\}$ between $\hat{p}$ and the mask proposal lists:

$$S = Softmax[sim(\hat{p}, v_1)/\tau, ..., sim(\hat{p}, v_M)/\tau], \tag{4}$$

then we multiplied these scores with the corresponding proposals to get a new combined heat map: $H = s_1 * v_1 + s_2 * v_2 + ...s_M * v_M$, where the region matching the subject has a larger weight. The utilization of H is introduced in section 3.5.

## 3.4 RETRIEVAL-AUGMENTED INSTANCE NORMALIZATION

As we can see from Figure 1 (a), divergences in the appearance, action mode, shape, and background distribution of the same type of objects in different domains are the main reason causing domain shift, making it difficult to associate the correct text-object region when the model trained in the source domain migrates to the target. Previous works Nam et al. (2021); Peng et al. (2022) use AdaIN to introduce statistics from other data within the same mini-batch, increasing the style diversity of samples while keeping their content unchanged, which can enhance the robustness of the model. However, the style introduced by these methods is limited to a mini-batch, and it is difficult to select appropriate data to effectively enhance the samples. Thus in this work, we propose Retrieval-augmented Instance Normalization (RaIN), which can use the powerful text-image matching ability of

the CLIP model to retrieve a series of similar objects that are most consistent with the semantics of the current questions, and extract statistical styles from them to enhance the current sample. Specifically, similar to Moco He et al. (2020), we maintain the dictionary $D$ as a queue of object mask proposals, the objects in the dictionary can be replaced progressively.

For a given visual feature, we first use CLIP text encoder to extract its counterpart query's feature as q. Then we use CLIP model to retrieve Top $k$ objects $\mathbf{O}_i = \{o_1^i, ..., o_k^i\}$ from $D$ based on their similarities with q. We use these similar objects to update each proposal $v_i$ within current object lists with Instance Normalization:

$$\hat{\mu} = (1-\lambda)\mu(v_i) + \lambda \cdot \mu(\mathbf{O}_i); \quad \hat{\sigma} = (1-\lambda)\sigma(v_i) + \lambda \cdot \sigma(\mathbf{O}_i); \quad \tilde{v}_i = \hat{\sigma}(\frac{v_i - \mu(v_i)}{\sigma(v_i)}) + \hat{\mu}, \ (5)$$

then we add $\{v_1, v_2, ...v_m\}$ into the dictionary $D$. After RaIN, we can obtain the enhanced visual feature $\tilde{V}$, we replace $\hat{V}$ with $\tilde{V}$ to do CRPrompt in Section 3.3: $\tilde{V} = \{v_{cls}, v, \tilde{v}_1, \tilde{v}_2, ...\tilde{v}_m\}$.

### 3.5 TRAINING

Following Hui et al. (2021), we use multi-scale encoder-decoder architecture as our backbone network, the shape in each scale layer can be denoted as $\hat{V}_k \in R^{H_k \times W_k \times C}(k \in \{1, K\})$, in this work, we set K as 5. We use two convolution layer to compress the heat map $H$ into the same size as $\hat{H} \in R^{H_2 \times W_2 \times C}$, and integrate $\hat{H}$ into $\hat{V}_2$ with a feed-forward layer: $\hat{V}_2 = FF(\hat{H} \oplus \hat{V}_2)$. We also introduce a feed-forward layer to integrate the query prompt $P = [P_1; P_2]$ into the Bert encoding result $\hat{Q}$. After training, we can obtain multi-scale response map $\{\{s_{i,j}^k\}_{i=1,j=1}^{H_i^k \times W_j^k}\}(k \in [1, K]))$ from the output of each decoder layer, and the ground-truth pixel-level annotations $\{\{y_{i,j}^k\}_{i=1,j=1}^{H_i^k \times W_j^k}\}(k \in [1, K]), y_{i,j}^k \in \{0, 1\})$, we can compute the multi-scale segment loss with binary cross-entropy loss:

$$\mathcal{L}_{\text{seg}} = -\frac{1}{K}\frac{1}{H_k W_k}\sum_{h=1}^{H_k}\sum_{w=1}^{W_k}(CE(y_{i,j}, s_{i,j})) \tag{6}$$

The total loss function is $\mathcal{L} = \mathcal{L}_{\text{seg}} + \mathcal{L}_{\text{const}}$.

## 4 EXPERIMENTS

### 4.1 DATASETS AND GENERALIZATION TASKS

**A2D Sentences** is first released by Gavrilyuk et al. (2018), they provide corresponding natural language descriptions for each video in Actor-Action Dataset Xu et al. (2015). It has a total of 3,782 videos, which contain 8 action classes and 7 actor classes (e.g. adult, dog, cat).

**Refer-Youtube-VOS (RVOS)** is first extended on the Youtube-VOS dataset by Seo et al. (2020), which contains 3975 high-resolution videos with 94 common object categories.

In this paper, we mainly study **Open-set Domain Generalization (OSDG)** task, we use the above-mentioned A2D and RVOS datasets to train the model on one dataset and then test its generalization ability on another dataset (**A2R** & **R2A**). The target domain has at least 3 object types that do not appear in the source domain. More details about A2R and R2A generalization settings can be found in appendix.

### 4.2 BASELINES AND EVALUATION METRICS

We compare our method with several state-of-the-art open-set referring segmentation models: CLIPSeg Lüddecke & Ecker (2022), ReCLIP Subramanian et al. (2022), ZSSeg Xu et al. (2021), and DG models: AdaIN Huang & Belongie (2017), Instance Selective Whitening (ISW) Choi et al. (2021). Following previous works, we adopt intersection-over-union (IoU) to measure the model segmentation ability, more implementation details can be found in the appendix.

Table 1: Comparison with state-of-the-art methods on open-set domain generalization tasks.

| Method | Precision | | | | | mAP | IoU | |
|---|---|---|---|---|---|---|---|---|
| | P@0.5 | P@0.6 | P@0.7 | P@0.8 | P@0.9 | 0.5:0.95 | Overall | Mean |
| R2A | | | | | | | | |
| Baseline | 50.45 | 43.04 | 32.59 | 19.12 | 3.02 | 26.60 | 52.46 | 44.26 |
| ISW | 52.25 | 45.09 | 34.59 | 21.51 | 4.53 | 28.95 | 54.32 | 46.11 |
| AdaIN | 54.18 | 47.52 | 37.92 | 23.01 | 4.92 | 30.71 | 56.30 | 48.08 |
| ZSSeg* | 54.83 | 48.13 | 39.00 | 23.99 | 5.05 | 31.39 | 56.03 | 48.05 |
| ReCLIP* | 55.01 | 48.80 | 39.36 | 24.22 | 5.53 | 31.75 | 56.83 | 48.52 |
| CLIPSeg* | 55.29 | 49.60 | 40.03 | 25.28 | 5.78 | 32.45 | 56.62 | 47.99 |
| CLIPSeg+AdaIN | 54.83 | 50.55 | 42.1 | 26.90 | 6.41 | 32.92 | 56.36 | 48.04 |
| ReCLIP+AdaIN | 56.47 | 50.76 | 42.37 | 26.77 | 6.31 | 33.61 | 56.45 | 48.98 |
| **Our Model** | **57.92** | **51.81** | **43.14** | **27.85** | **7.18** | **34.66** | **57.87** | **49.94** |
| A2R | | | | | | | | |
| Baseline | 35.02 | 28.18 | 19.70 | 10.24 | 1.62 | 17.04 | 38.95 | 33.30 |
| ISW | 36.14 | 28.96 | 19.98 | 10.99 | 1.87 | 17.66 | 39.71 | 34.12 |
| AdaIN | 37.04 | 30.09 | 22.06 | 11.59 | 2.27 | 17.99 | 41.27 | 35.13 |
| ZSSeg* | 38.04 | 30.77 | 21.87 | 10.99 | 2.41 | 18.92 | 41.50 | 35.24 |
| ReCLIP* | 37.10 | 30.71 | 22.41 | 12.53 | 2.68 | 19.17 | 41.41 | 35.26 |
| CLIPSeg* | 37.79 | 31.46 | 23.03 | 12.92 | 2.79 | 19.63 | 41.46 | 35.65 |
| ReCLIP+AdaIN | 38.04 | 31.19 | 22.89 | 12.55 | 3.04 | 19.55 | 41.89 | 35.85 |
| CLIPSeg+AdaIN | 38.56 | 31.65 | 22.74 | 12.23 | 3.20 | 19.64 | 41.93 | 36.07 |
| **Our Model** | **39.12** | **32.08** | **23.42** | **13.32** | **3.56** | **20.47** | **42.79** | **36.61** |

## 4.3 MAIN RESULTS

We compare our proposed approach with a series of state-of-the-art open-set referring segmentation models and domain generalization methods on two datasets. We reimplement these state-of-the-art methods on the new proposed task. The main evaluation results are presented in Table 1. From the results we can see that our method achieves remarkable performance gains of about 4∼7% than the baseline model on two generalization directions, demonstrating the effectiveness of our approach. Using the DG method alone can slightly improve the performance of the model in unknown domains, but the improvement is less obvious as using the CLIP-based open-set segmentation methods. We think there are two main reasons: 1. in this task, the label shift has a higher impact on the model than the domain shift. 2. the introduction of CLIP can help the model resist some domain shifts. The results also show that combining the two kinds of methods together can improve the performance of the model. However, our model can still outperform these methods, which further demonstrates that our model can transfer the segmentation ability to unknown objects by reasoning from seen vision-language knowledge.

## 4.4 ABLATION STUDY

**Effectiveness of CRPrompt.** The CRPrompt contains two different ways of reasoning, implicit reasoning (IR) and explicit reasoning (ER). We remove these two reasoning modules respectively to study their effects. As illustrated in Table 2, both of the two reasoning modules can improve the generalization ability of the model, which significantly increases the Overall IoU accuracy of 2.70% and 3.09% in A2R, 1.84% and 2.01% in R2A. The results of ER demonstrate that the introduction of CLIP can effectively link unknown words and visual regions together, which can also significantly improve the open-set ability of the model. Besides, the performance gained by introducing IR proves that our method can effectively use CLIP to extract seen text-visual paired knowledge as prompt to facilitate unknown objects segmentation, which is also an important difference between our model and other CLIP-based open-set segmentation methods.

**Effectiveness of Prompt Engineering.** To further evaluate the performance of the prompt engineering, we remove the text prompt and visual prompt respectively, as shown in Table 3. The results illustrate that the performances degrade without the two kinds of prompts. The reason is that the text prompt can allow the model to distinguish between the main object to be segmented and other

Table 2: Analysis of the components on two generalization tasks.

| Method | Precision | | | | | mAP | IoU | |
|---|---|---|---|---|---|---|---|---|
| | P@0.5 | P@0.6 | P@0.7 | P@0.8 | P@0.9 | 0.5:0.95 | Overall | Mean |
| **A2R** | | | | | | | | |
| *Full Model* | **39.12** | **32.08** | **23.42** | **13.32** | **3.56** | **20.47** | **42.79** | **36.61** |
| w/o Implicit Reasoning | 36.68 | 29.34 | 21.66 | 11.36 | 2.52 | 18.39 | 40.09 | 34.20 |
| w/o Explicit Reasoning | 35.96 | 29.75 | 20.37 | 11.64 | 2.43 | 18.04 | 39.71 | 34.16 |
| w/o RaIN | 37.10 | 30.71 | 22.41 | 12.53 | 2.68 | 19.17 | 41.41 | 35.26 |
| RaIN→AdaIN | 38.10 | 31.86 | 22.91 | 12.28 | 2.89 | 19.54 | 40.91 | 35.52 |
| Baseline | 35.02 | 28.18 | 19.70 | 10.24 | 1.62 | 17.04 | 38.95 | 33.30 |
| **R2A** | | | | | | | | |
| *Full Model* | **57.92** | **51.81** | **43.14** | **27.85** | **7.18** | **34.66** | **57.87** | **49.94** |
| w/o Implicit Reasoning | 55.01 | 48.67 | 40.18 | 25.15 | 5.51 | 32.14 | 56.03 | 47.81 |
| w/o Explicit Reasoning | 54.10 | 48.03 | 39.26 | 24.31 | 5.13 | 31.51 | 55.86 | 47.29 |
| w/o RatIN | 55.98 | 49.37 | 40.64 | 25.20 | 6.02 | 32.14 | 56.03 | 47.81 |
| RatIN→AdaIN | 56.29 | 49.99 | 41.13 | 26.33 | 6.74 | 33.26 | 55.09 | 48.99 |
| Baseline | 41.72 | 33.59 | 24.23 | 12.81 | 2.66 | 20.88 | 42.64 | 37.67 |

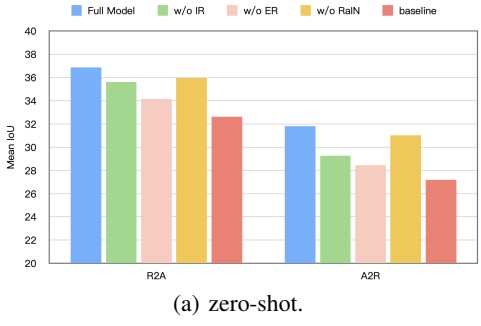

(a) zero-shot.

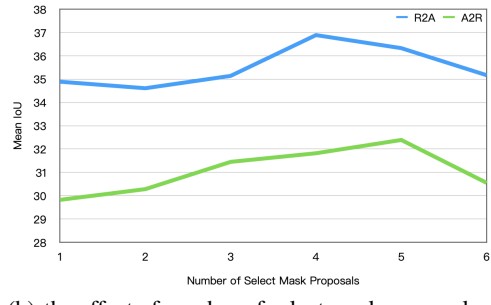

(b) the effect of number of select mask proposals.

Figure 3: The effect of different codebook sizes on two pre-training tasks.

objects which can provide contextual clues. And the visual prompt can help the model improve its generalization ability in unknown objects. By introducing text and visual prompts, our model can facilitate CLIP to achieve more precise text-object matching, surpassing previous CLIP-based open-set referring segmentation methods.

**Effectiveness of RaIN.** We conduct a series of experiments to test the performance of our proposed RaIN. As we can see in Table 2, the accuracy will decrease significantly at all levels. To further illustrate the superiority of our module, we replace RaIN with the widely used AdaIN. From Table 2 we can observe that although AdaIN can help our model improve performance in unknown domains, it is not as effective as RaIN. Compared with AdaIN using data in the same batch, RaIN can introduce object statistics with similar semantics to enhance the current objects, simulating their styles in unknown domains, which can accurately improve the generalization ability of the model.

**Effectiveness of The Selection Number of Mask Proposals.** We also test the performances when the selection number $M$ of mask proposals is set to a different number. From Figure 3 (b) we can see that the model performs best when $M = 4$ in R2A, and when $M = 5$ in A2R. Some main objects may not be selected if $M$ is too small, while the text-region matching ability will be reduced if $M$ is too large. Therefore, considering performance and efficiency, we set M to 4 in this paper.

## 4.5 PERFORMANCE OF ZERO-SHOT DOMAIN GENERALIZATION

The above experiments prove that our proposed modules can effectively improve the performance of the model under open-set setting. However, it is still unknown whether the improvement mainly comes from seen objects or unseen objects in target domains. To illustrate this, we conduct a series

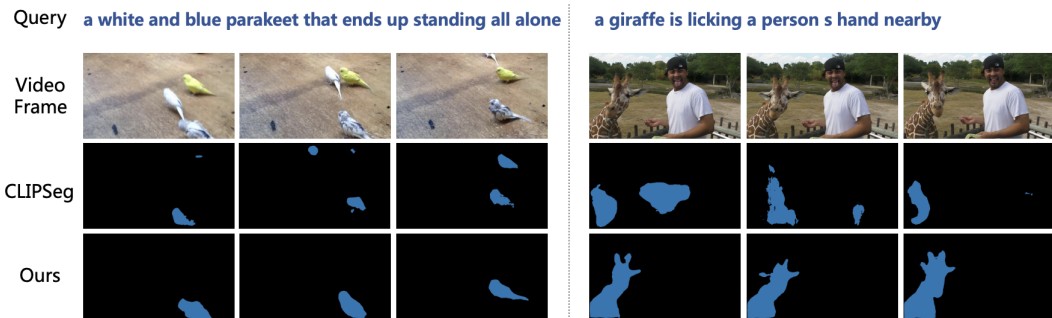

Figure 4: The visualization of segmentation results. Our model is better than CLIPSeg in segmenting unseen categories.

Table 3: Analysis of the prompt engineering in CRPrompt.

| Method | A2R | | | R2A | | |
| --- | --- | --- | --- | --- | --- | --- |
| | mAP 0.5:0.95 | IoU Overall | Mean | mAP 0.5:0.95 | IoU Overall | Mean |
| *Full Model* | **20.47** | **42.79** | **36.61** | **34.66** | **57.87** | **49.94** |
| w/o Text Prompt | 19.55 | 41.29 | 35.21 | 33.44 | 56.87 | 48.44 |
| w/o Visual Prompt | 19.03 | 41.01 | 34.75 | 31.36 | 54.92 | 46.37 |

of experiments under zero-shot domain generalization setting, where all objects in the target domain are unseen in the source domain. As shown in Figure 3 (a), we can see that the performances degrade dramatically without Implicit Reasoning (IR) and Explicit Reasoning (ER) in both two tasks, demonstrating our proposed CRPrompt plays an essential role in unknown objects segmentation. Meanwhile. we can observe that the removal of RaIN has little effect on the zero-shot capability of the model, which may be because RaIN focuses more on the robustness of seen object categories.

### 4.6 QUALITATIVE ANALYSIS

To qualitatively analyze the effectiveness of our method, we visualized two segmentation results on OSDG task, as shown in Fig 4. From the results we can see that compared with CLIPSeg model, our model can locate the object to be recognized and segment it accurately. The right demo can show that: although our model has not seen the object "giraffe" in the source domain before, it can still predict the right region of giraffe based on the learned knowledge from seen object "people" and the spatial location relationship "nearby". The demo on the left proves that even if there are no seen objects as a reference in the video frames, the model can still determine which one is the right parrot to be recognized based on knowledge such as color and spatial relationship learned from the source domain, while the CLIPSeg model can seldom tell the right parrot.

## 5 CONCLUSION

In this paper, we investigate a challenging problem, open-set domain generalization in referring video segmentation task, where the model is required not only to recognize unknown objects, but also to be able to segment them according to the text description, overcoming both domain shift and label shift. To migrate the huge gap between the source domain and target domain, we bring up CRPrompt, which can extract the learned knowledge from the source domain as text and visual prompt, to help CLIP model achieve better text-region alignment, and transfer the segmentation ability from seen objects to unseen objects. Furthermore, we propose a RaIN to reduce the domain shift caused by different distributions of objects in different domains such as appearance, shape, background and action mode. Extensive ablation studies on open-set and zero-shot domain generalization tasks verify the effectiveness of each proposed component.

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

## A    APPENDIX

## B    DATASETS DETAILS

In open-set domain generalization setting, we choose **"person, parrot, dog, cat, skateboard, giraffe, motorbike, duck, mouse, giant panda"** categories in RVOS dataset, and choose **"adult, baby, ball, bird, cat, dog, car"** categories in A2D dataset. In zero-shot domain generalization setting, we choose the same categories as OSDG settings in A2D dataset, and we choose all the categories except "adult, baby, ball, bird, cat, dog, car" in RVOS dataset.

## C    EVALUATION METRICS

For IoU, we use the "Overall IoU", which calculates the ratio of the total intersection area divided by the total union area over the entire dataset, and the "Mean IoU", which first calculates the ratio of each sample and then obtains the average results on the whole dataset. "P@K" denotes compared with ground truth results, the IoU scores of testing samples are larger than K. We also measure the mean average precision at 5 different IoU thresholds from 0.50 to 0.95 with the step 0.05.

## D    IMPLEMENTATION DETAILS

For natural language query inputs, we set the maximum number of words in one query as 20, and apply the BertDevlin et al. (2018) as our text encoder. For video inputs, we employ the I3D network Carreira & Zisserman (2017) pretrained on the Kinetics dataset to extract the spatial and temporal features and we use the pre-trained ResNet-50 He et al. (2016) to extract each video frame representations, the number of frames in one clip is 8. We select $M = 4$ mask proposals using Mask-RCNN for each video frame, then we resize these proposals to CLIP resolution, and use "ViT-B/32" as the visual encoder for proposals. We also use the CLIP text encoder to complete text prompt in CRPrompt module.

We divide the visual features into $K = 5$ different scales, the sizes of them are $320 \times 320$, $160 \times 160$, $80 \times 80$, $40 \times 40$ and $20 \times 20$ separately. We set the hidden size of visual and query features as 512. Following Wang et al. (2019), the FCN network in deconvolutional layer contains three fully convolutional layers, where the kernel size is 3×3 for the first two layers and 1×1 for the remaining layer. All experiments are implemented with Pytorch package on 4 NVIDIA V100 GPUs in this paper, the batch size is 16, and we use Adam optimizer with a initial learning rate 1e-7.

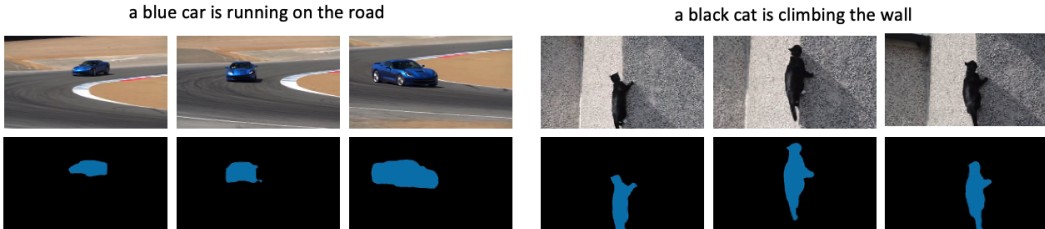

Figure 5: The visualization results of open-set domain generalization task:**R2A**.

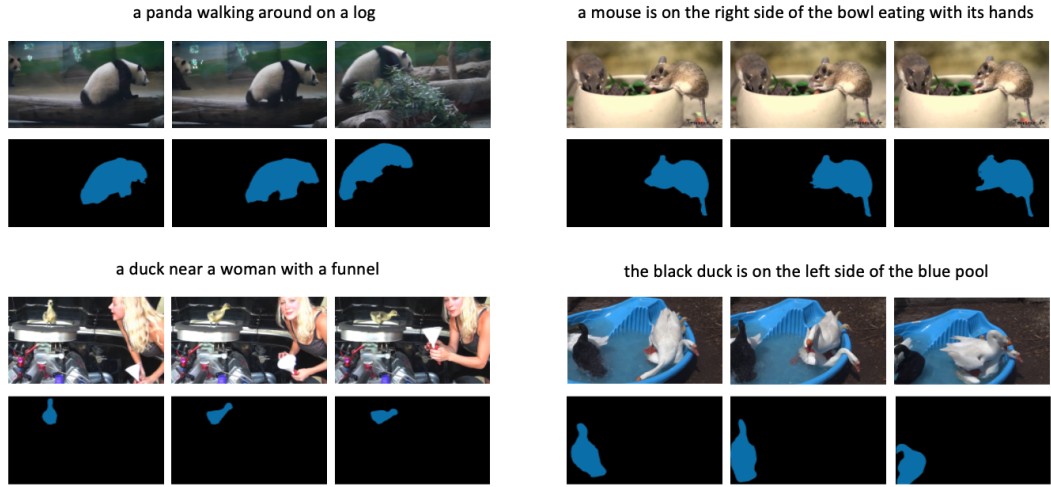

Figure 6: The visualization results of open-set domain generalization task:**A2R**.

## E  TEXT PROMPT DETAILS

Here we use an example to give a detailed description of the text prompt. Given a query *"a motor bike being used by a man riding on a dirt road in front of another man riding a motorbike"*, we can use spaCy Honnibal & Johnson (2015) to split it into several phrase fragments: *"a motor bike/ being used by a man/ riding on a dirt road/ in front of/ another man riding a motorbike"*. The first fragment *"a motor bike"* is the subject part $Q_1$ to be segmented, and the following parts $Q_2$ are clues to locate it. We use "and" to connect the fragments in $Q_2$. Then we introduce natural language prompts to modify $Q_1$ and $Q_2$: $\hat{Q}_1 = [L_1; Q_1; L_2]$, thus $\hat{Q}_1$ is "***Localizing** a motor bike, **which is the main object to be segmented.**"; $\hat{Q}_2 = [L_3; Q_2]$, thus $\hat{Q}_2$ is "**Find clues to locate the previously mentioned subject from the following objects and relationships,** being used by a man and riging on a dirt road and in front of and another man riding a motorbike.".*

## F  MORE QUALITATIVE RESULTS ON TWO GENERALIZATION TASKS.

We show more qualitative results on two open-set domain generalization tasks, R2A (Figure 5) and A2R (Figure 6).

## G  REMARK

As illustrated in Figure 7, our model uses two steps to gradually minimize the distance between the source domain $\mathcal{S}$ and the target unknown domain $\mathcal{T}_u$. First, we use the proposed SelectIN to constraint the distribution divergence between $\mathcal{S}$ and $\mathcal{T}_k$: $D\left(\mathcal{S}, \mathcal{T}_k\right) \leq \rho_1$. Then we use the knowledge learned from $\mathcal{S}$ to facilitate CLIP model to align the unknown words $Q_u$ and unseen visual regions $V_u$. The seen objects $Q_k$ and $V_k$ in $\mathcal{T}_k$, and the large-scale text-image paired knowledge existing in CLIP,

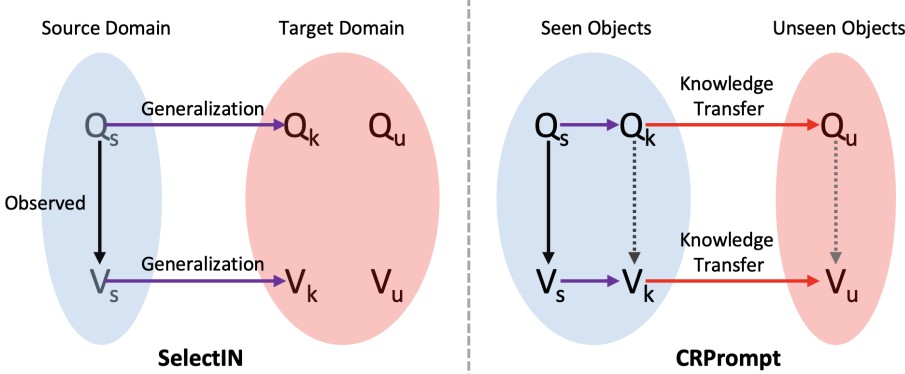

Figure 7: The SelectIN can align the distribution between $\mathcal{S}$ and $\mathcal{T}_k$, then the CRPrompt can transfer knowledge learned from seen objects in $\mathcal{S}$ and $\mathcal{T}_k$ to unseen objects in $\mathcal{T}_u$.

can serve as a bridge to transfer the segmentation ability from $\mathcal{S}$ to $\mathcal{T}_k$. Thus we can minimize the distance between $\mathcal{S}$ and $\mathcal{T}_k$: $D\left(\mathcal{S}, \mathcal{T}_u\right) \leq \left[D\left(\mathcal{S}, \mathcal{T}_k\right) + D\left(\mathcal{T}_k, \mathcal{T}_u\right)\right] \leq \left(\rho_1 + \rho_2\right)$.

