# OpenReview forum: "Overcoming both Domain Shift and Label Shift for Referring Video Segmentation"
_ICLR.cc/2024/Conference — Submitted to ICLR 2024_

### Official Review · Reviewer_qcvG · 2023-10-28

**Soundness:** 2 fair
**Presentation:** 2 fair
**Contribution:** 2 fair
**Rating:** 3
**Confidence:** 4

**Summary:**

This paper focuses on Open-Set Domain Generalization (OSDG) task and aims to address the domain and label shifts for RVOS. To tackle the label shift challenge, the authors propose CLIP-based Reasoning Prompt (CRPrompt), leveraging combined textual and visual prompts to enhance CLIP's text-object matching ability. Additionally, a RaIN operation is introduced to enhance the robustness of models in the OSDG setting. Experimental results are conducted on open-set and zero-shot domain generalization tasks.

**Strengths:**

(1)	The paper explores two shift problems exists in RVOS task, including domain shift and label shift, which is new for the RVOS methods.
(2)	The idea of combining visual and textual prompts in CRPrompt for prompt engineering is reasonable for referring video segmentation.

**Weaknesses:**

(1)	The authors claim their paper introduces the first multimodal Open-set Domain Generalization (OSDG) method, which overclaims on its contribution since compared with RVOS, other text-image multimodal tasks RES, REC, image captioning, and VQA are more straightforward to carry on this research. Besides, the proposed method does not make any video-relevant improvements either.
(2)	the paper tackles the challenges of domain shift and category shift, but these challenges are not entirely isolated. For instance, category shift can inherently cause domain shift to some extent. The authors present contributions for each challenge separately but lack a profound exploration of the interrelation between these issues. This oversight diminishes the paper's academic rigor, making it resemble a technical report rather than a scholarly contribution.
(3)	The experimental comparisons made in the paper are unfair. The authors compare their method with other OV methods which have no domain generalization capabilities on the RVOS dataset. Moreover, they only combine these OV methods with an older Domain Generalization (DG) method, AdaIN, for comparison against their own approach. This method of comparison is clearly unfair. A fair comparison should involve combining these methods with updated DG methods or the proposed RaIN method to showcase the advantages of the paper's approach. Furthermore, the paper lacks results from supervised learning on the same dataset, which would provide a valuable benchmark for evaluating the effectiveness of their OSDG method.

**Questions:**

See weaknesses.

---

### Official Review · Reviewer_hGmE · 2023-10-30

**Soundness:** 3 good
**Presentation:** 3 good
**Contribution:** 3 good
**Rating:** 6
**Confidence:** 3

**Summary:**

This paper explores the label shift and domain shift issues of OSGD on the task of referring video segmentation, and further explores how to make the model segment unseen categories. This paper proposed CRPrompt and RaIN to solve the above problems and proved their effectiveness.

**Strengths:**

1.	The research question is important. Compared with the previous OSDG, this paper further explores how to make the model segment unseen categories. The model can make use of the known objects around and learned knowledge to predict unseen targets.
2.	The paper conducts experiments on domain generalization and zero-shot and proves the effectiveness of their methods.

**Weaknesses:**

1.	The author only describes CRPrompt in Figure 2, but there should be a diagram to describe the entire model, including CRPrompt, RaIN and segmentation, to express their relationship and which parts participate in training.
2.	The method in this article seems to be dependent on the performance of the model that generates Mask Proposals, as well as the models that generate Q1 and Q2.
3.	Some parts are not described clearly, such as in 3.5, why integrate H only into V2 instead of V1 ，V2… Vk

**Questions:**

1.	In section 3.5, why integrate H only into V2 instead of V1 ，V2… Vk
2.	In formula（2）, what is the meaning of introducing visual prompt Pv into P1 and P2 at the same time?
3.	In formula (3), N is the batch size, and Vcls is not related to n. Does this mean that Vcls is an embedding obtained on the entire batch size? If so, does it mean that C1...Cn in Figure 2(a) are the same?

---

### Official Review · Reviewer_GHiP · 2023-11-01

**Soundness:** 2 fair
**Presentation:** 2 fair
**Contribution:** 2 fair
**Rating:** 5
**Confidence:** 3

**Summary:**

This paper is aimed to improve model robustness against domain shift and label shift in open-set domain generalization problems. It focuses on video segmentation and introduces CLIP-based Reasoning Prompt, leveraging CLIP to enhance object segmentation for unknown classes using text-visual prompts. Additionally, Retrieval-augmented Instance Normalization enhances the model by retrieving semantically similar objects. Cross-dataset experiments validate the approach's effectiveness.

**Strengths:**

- To leverage CLIP to enhance object segmentation for unknown classes, prompt engineering based on implicit reasoning and explicit
reasoning are introduced.
- Cross-dataset experiments of A2R & R2A prove the effectiveness of the proposed approach.

**Weaknesses:**

- The authors claim this work is the first attempt to solve open-set domain generalization problem on multi-modal task, while, the experiments are evidenced by the cross-dataset experiments of A2R & R2A. This cross-dataset setting is widely used in previous open-vocabulary/set/world papers. I am very confused why the authors claim this paper is the first work ?
- Following, I am curious about how the authors define the differences between open-set domain generalization and open-vocabulary problems, since CLIPSeg is an open-vocabulary segmentation method In Table 1.
- The proposed CLIP-based Reasoning Prompt seems a general method for object-level perception task, for example, object detection and segmentation. Why do the authors choose referring video segmentation as the task ?

My initial rating is marginally below the acceptance threshold, but I'm willing to increase the score if my confusions are well addressed.

**Questions:**

- Intuitively, retrieval-augmented Instance Normalization has little with the problem open-set domain generalization. Although it is able to improve the performance, it is more like an implementation detail, instead of technical contribution.

---

### Official Review · Reviewer_WCGM · 2023-11-09

**Soundness:** 2 fair
**Presentation:** 1 poor
**Contribution:** 1 poor
**Rating:** 3
**Confidence:** 4

**Summary:**

This paper tackles the referring video object segmentation task (RVOS) with a special focus on open-set scenarios, which have been one of the huge interests of the community.
The authors adopt multiple recent findings from previous works, and present CRPrompt.
Specifically, CRPrompt leverages both nlp and visual queries and by adding some prompt engineerings with cross attentions, the authors aim to drive better representations within queries.
By improving the model using implicit reasoning, explicit reasoning, and Retrieval-augmented Instance Normalization (RaIN), this paper achieves high accuracies on Open-set Domain Generalization (OSDG) tasks.

**Strengths:**

The open-set problems is one of the main problems that the community recently have been aiming to improve.
This paper presents multiple recent works that are suitable to use for improving the overall accuracy.
Each module that they adopt or present benefits the performance, and the authors prove these positive aspects by achieving high accuracies on OSDG tasks.

**Weaknesses:**

Motivation
- It is somewhat not intuitive why the video task is used to prove the effectiveness of each module. Specifically, the modules that the authors present can be applied to the image domain, and I believe there aren't much considerations on the "video" aspects. For instance, none of the video-specific failure cases such as occlusions and consistency are dealt in this paper. If it could be directly applied to the image domain, which is a bit more major, why did the authors aimed to target the video domain?

Main contribution
- To my understanding, the claimed contributed features of this paper are implicit reasoning, explicit reasoning and RaIN. However, there are too many components that are either added or modified from baselines, e.g. backbone is changed to I3D. However, there are limited ablations that actually show which part of the model actually benefits the accuracy.
- I appreciate the authors for improving the accuracy by the proposed method. However, the contributions seem marginal to me, with some slight changes or applications of existing literatures. Instead of sharing only the accuracy drops in Table 2, I personally believe it could be much more insightful if actual analysis (besides the accuracy comparisons) were provided.

Writing
- Overall, I believe the writing is not clear enough to easily follow. I believe this is mostly due to unnecessary mathematical expressions that  can be briefly explained.
- What is D and rho in eq 1? Probably D signifies distances, but these should be explained.

**Questions:**

- What's the FLOPs of this method compared to other works? Do they have much higher FLOPs due to adopting different backbone?

- Please share more details on the adopted methods. For example,
    - Which dataset is MaskRCNN trained on? What if it fails to detect necessary objects?
    - What are the backbones that MaskRCNN use? What size is the I3D backbone? More details on GRU module?

Please also refer to the weaknesses section.

---

### Meta-Review · Area_Chair_1r4U · 2023-12-05

**Metareview:**

The ratings are 1 borderline accept, 1 borderline reject, and 2 reject. The main concerns from the reviewers are: 1) technical clarity to describe more details about the proposed method; 2) missing context about the problem setting for open-set domain generalization; 3) experimental settings that may not be fair. Since no rebuttal is provided to address these issues, the rejection rating is recommended.

**Justification For Why Not Higher Score:**

The rebuttal was not provided from the authors to address the concerns raised from the reviewers.

**Justification For Why Not Lower Score:**

N/A

---

### Decision · Program_Chairs · 2024-01-16

Reject